# Cardioprotective Effects of Dexmedetomidine in an Oxidative-Stress In Vitro Model of Neonatal Rat Cardiomyocytes

**DOI:** 10.3390/antiox12061206

**Published:** 2023-06-02

**Authors:** Moritz Borger, Clarissa von Haefen, Christoph Bührer, Stefanie Endesfelder

**Affiliations:** 1Department of Neonatology, Charité—Universitätsmedizin Berlin, 13353 Berlin, Germany; moritz.borger@charite.de (M.B.); christoph.buehrer@charite.de (C.B.); 2Department of Anesthesiology and Intensive Care Medicine, Charité−Universitätsmedizin Berlin, 13353 Berlin, Germany; clarissa.von-haefen@charite.de

**Keywords:** hyperoxia, hypoxia, oxidative stress, preterm heart, dexmedetomidine, cardiomyocytes

## Abstract

Preterm birth is a risk factor for cardiometabolic disease. The preterm heart before terminal differentiation is in a phase that is crucial for the number and structure of cardiomyocytes in further development, with adverse effects of hypoxic and hyperoxic events. Pharmacological intervention could attenuate the negative effects of oxygen. Dexmedetomidine (DEX) is an α2-adrenoceptor agonist and has been mentioned in connection with cardio-protective benefits. In this study, H9c2 myocytes and primary fetal rat cardiomyocytes (NRCM) were cultured for 24 h under hypoxic condition (5% O_2_), corresponding to fetal physioxia (pO_2_ 32–45 mmHg), ambient oxygen (21% O_2_, pO_2_ ~150 mmHg), or hyperoxic conditions (80% O_2_, pO_2_ ~300 mmHg). Subsequently, the effects of DEX preconditioning (0.1 µM, 1 µM, 10 µM) were analyzed. Modulated oxygen tension reduced both proliferating cardiomyocytes and transcripts (CycD2). High-oxygen tension induced hypertrophy in H9c2 cells. Cell-death-associated transcripts for caspase-dependent apoptosis (Casp3/8) increased, whereas caspase-independent transcripts (AIF) increased in H9c2 cells and decreased in NRCMs. Autophagy-related mediators (Atg5/12) were induced in H9c2 under both oxygen conditions, whereas they were downregulated in NRCMs. DEX preconditioning protected H9c2 and NRCMs from oxidative stress through inhibition of transcription of the oxidative stress marker GCLC, and inhibited the transcription of both the redox-sensitive transcription factors Nrf2 under hyperoxia and Hif1α under hypoxia. In addition, DEX normalized the gene expression of Hippo-pathway mediators (YAP1, Tead1, Lats2, Cul7) that exhibited abnormalities due to differential oxygen tensions compared with normoxia, suggesting that DEX modulates the activation of the Hippo pathway. This, in the context of the protective impact of redox-sensitive factors, may provide a possible rationale for the cardio-protective effects of DEX in oxygen-modulated requirements on survival-promoting transcripts of immortalized and fetal cardiomyocytes.

## 1. Introduction

Improved perinatal and neonatal care of preterm infants has yielded higher survival rates, especially for very and extremely preterm infants [1]. The first preterm infants to benefit from the care-related medical improvements are now 30 to 40 years old. The primary focus, with interventions in the first years of life to reduce neonatal and early childhood mortality and morbidity, inevitably leads to a secondary focus on the long-term health in adulthood of the prematurely born. It is already known that prematurity can lead to longer-term impairments. These long-term sequelae include motor and cognitive deficits [2], impaired psychological morbidities and socio-emotional behavioral disorders, such as attention deficit hyperactivity disorder (ADHD) and autism spectrum disorders (ASDs) [3,4,5], as well as higher levels of cardiometabolic risk factors as adults [6]. These risk factors involve hypertension [7], impaired glucose tolerance and regulation [8,9], obesity [10], and hyperlipidemia [6].

In this regard, it can be observed that preterm birth is associated with a two-fold increase in the risk of death from cardiovascular causes with an inverse association to birth weight throughout adolescence and adulthood [11], and impaired exercise tolerance [12]. Clinical and experimental evidence suggests that preterm birth is associated with impaired or disturbed structural or functional cardiovascular and pulmonary development [13]. Little is known about the etiology; however, structural and functional alterations of the premature heart that could favor the progress of cardiovascular malfunctions have been described. Specifically, smaller left and right ventricular volumes with greater ventricular masses were observed in prematurely born young adults. Their hearts were more likely to present diffuse fibrosis and hypertrophic cardiomyocyte growth. These changes are, amongst other functional alterations, accompanied by a reduced right ventricular ejection fraction and a compromised left ventricular ejection fraction under moderate and high intensity exercise, indicating an impaired capacity of circulatory adaptation in people born preterm (reviewed by [14]). During birth, the fetus is exposed to an environment of relative hyperoxia, causing the partial oxygen tension to rise from approximately 25–35 mmHg in the umbilical vein and 15–25 mmHg in the umbilical artery to 40–80 mmHg in the arterial blood of the newborn [15]. Organ maturation is particularly advanced in the last trimester of pregnancy, which must be continued ex utero in the case of preterm birth. Oxygen conditions change abruptly, to which preterm infants have problems adapting. Oxidative stress, based on different mechanisms [16,17], leads to increased levels of free radicals. With an immature antioxidant enzyme system in premature infants [18], this subsequently leads to oxidative damage to organs [19]. Relative hyperoxia predisposes to the formation of free radicals and can be exacerbated by the need for oxygen supplementation as well as additional mechanical ventilatory support, further increasing both oxidative stress and the oxidative stress response. Cell cultures are routinely grown in incubators with O_2_ levels stabilizing at 18–19%. These cells are subjected to per se hyperoxia compared to 5% O_2_ grown cell (physioxia, i.e., equivalent to tissue normoxia). If the oxygen level is further increased, it is called extreme or absolute hyperoxia. Oxidative stress is defined as an imbalance between oxidants and antioxidants in favor of oxidants, potentially leading to damage [20]. Thus, the homeostasis of oxidative/antioxidative signaling pathways is disturbed, which has implications for downstream-mediated signaling pathways via redox-sensitive transcription factors [21]. Consequently, oxidative stress mainly caused by free radicals and the production of reactive oxygen species (ROS) generated during the reactions of the electron transport chain increases postnatally, exerting adverse effects on the integrity of cellular macromolecules [22,23]. Cardiomyocyte-associated factors of perinatal maturation have not yet been fully identified. As described, birth is associated with widespread changes in oxygenation, hemodynamics, and the surrounding biochemical milieu [24]. The transition from fetal to postnatal circulation imposes structural and contractile adaptations [25]. Adaptation from the hypoxic to the hyperoxic oxygen environment leads to a decrease in HIF1α regulation [26]. Impairments of this redox-sensitive process condition delayed cardiomyocyte maturation. Alteration of the oxygen environment too early, as in premature birth, leads to abnormal cardiomyocyte hypertrophy and premature cell cycle arrest both in the immature preterm infant and in diverse animal models [27,28,29].

Dexmedetomidine (DEX) is a highly selective α2-adrenoceptor agonist. Its application in pediatric and neonatal care has increasingly been investigated over the last decade, as DEX provides shorter ventilation durations and a reduced incidence of sepsis with no difference in sedation quality compared to standard therapies [30,31,32]. Various studies describe beneficial effects of DEX on various organ systems via antiapoptotic, anti-inflammatory and antioxidative mechanisms [33,34,35,36,37]. In rodent hearts, DEX administration attenuates hypoxia reoxygenation injury [38,39]. Furthermore, it was shown that DEX ameliorates oxidative-stress-induced apoptosis in neonatal rat cardiomyocytes, suggesting a role in cytoprotective antioxidative mechanisms [40,41]

This in vitro study investigates the cardio protective effect of dexmedetomidine on oxidative-stress-damaged immortalized and primary rat cardiomyocytes.

## 2. Materials and Methods

### 2.1. Cell Culture: Cell Line H9c2

Embryonic rat-heart-derived H9c2-myoblasts obtained from America Tissue Type Collection (Manassas, VA, USA, cat. CRL-1446) were cultured using Dulbecco’s Modified Eagle’s Medium (DMEM; Bio&SELL, Feucht/Nürnberg, Germany, cat. BS.11971083) supplemented with 10% fetal bovine serum, 100 U/mL penicillin and 100 μg/mL streptomycin (Bio&SELL, Feucht/Nürnberg, Germany, cat. BS.4681080) and grown at 37 °C in a 5% CO_2_ humidified atmosphere. The medium was changed every 3–4 days, and cells were subcultured when at 70–80% confluence. Before treatment, H9c2 cells were seeded in a density of 2.1 × 10^4^ cells/cm^2^ in 2 mL complete DMEM medium and grown for 24 h.

### 2.2. Cell Culture: Primary Neonatal Rat Cardiomyocytes (NRCM)

All experimental procedures were performed following institutional and international guidelines and were approved by the local animal welfare authorities (LAGeSo, Berlin, Germany, T-CH0019/20). Timed-pregnant Wistar rat dams were obtained through Janvier Labs (Le Genest-Saint-Isle, France) and housed in individual cages under controlled environmental conditions with ad libitum access to food and water. At embryonic day 18 (E18), adult rats were anesthetized with isoflurane and sacrificed. Pups were segregated from the uterus and transferred into ice-cold phosphate-buffered saline. Hearts were removed and cells were isolated using the Pierce Cardiomyocyte Isolation Kit (Thermo Scientific, Rockford, IL, USA, cat. 88281) following the manufacturer’s instructions. Cells were plated on 35 mm Petri dishes in a density of approximately 2.5x10^5^ cells/cm^2^ and grown in DMEM (Thermo Scientific) supplemented with 10% fetal bovine serum, 100 U/mL penicillin and 100 μg/mL streptomycin (Bio&SELL) at 37 °C in a 5% CO_2_ humidified atmosphere for 7 days. The medium was changed every 2–3 days.

### 2.3. Cellular Model and Defining Oxygen Level

To recognize the generated results and their distinction of the used oxygen concentrations for the relevance of the cell culture models, it must be mentioned that the use of the terms “hypoxia” (5% O_2_), “normoxia” (21% O_2_), and “hyperoxia” (80% O_2_) was useful here because of the simplified reference to the oxygen concentrations. The oxygen data refer to the composition of the gas space of the humidified incubator and do not correspond to the actual cellular oxygen environment of the cells. This is influenced by multiple factors, such as cell density, medium layer thickness, as well as incubation time [42]. Under these conditions, at an O_2_ content of 21%, the oxygen tension or oxygen partial pressure (pO_2_) is ∼150 mmHg and “normoxia” is accordingly used synonymously for atmospheric oxygen pressure [43]. Under physiological conditions, tissue pO_2_ ranges between 3% and 8% O_2_ (usually pO_2_ 23–70 mmHg) and is often referred to as “physioxia”; defined here as “hypoxia” in experimental terminology in relation to associated Hif1α targeting under low-oxygen conditions. The hypoxic condition fits with the fetal oxygen tension. The fetal pO_2_ is approximately ~45 mmHg at mid-gestation and ~32 mmHg closer to term [44]. Correlating with their gestational age, extremely premature infants show large variations in pO_2_ measurements in the first days of life, which may well display maximum values >150 mmHg [45]. “Hyperoxia” of cardiomyocytes in the experimental setting at 80% O_2_ with a pO_2_ ~300 mmHg is used to simulate extreme oxygen tensions.

### 2.4. Oxygen Exposure and Drug Administration

In accordance with the experimental conditions, H9c2 cells and primary NRCMs were randomly assigned to three groups: hypoxic environmental conditions (5% O_2_, hypoxia), atmospheric air (21% O_2_, normoxia), or hyperoxic environmental conditions (80% O_2_, hyperoxia). These groups were each divided into 4 experimental groups using (1) pure medium (control group), (2) medium with 0.1 μM dexmedetomidine (DEX; EVER Valinject GmbH, Unterach, Austria), (3) medium with 1 μM DEX, and (4) medium with 10 μM DEX, resulting in 12 experimental groups. Fresh medium with or without DEX was applied, and after a 2 h preincubation under normoxic conditions, cells were kept under 5%, 21% or 80% O_2_ at 37 °C in a humidified 5% CO_2_ atmosphere for 24 h.

### 2.5. RNA Extraction and qPCR

As previously described [46], total RNA was isolated by acidic phenol/chloroform extraction using RNA Solv Reagent (Omega Bio-Tek, Norcross, GA, USA, cat. R6830) following the manufacturer’s instructions. Total RNA was reverse transcribed and DNAse treated. The PCR products of genes of interest were quantified in real time using qPCRBIO Probe Mix Lo-ROX (PCR Biosystems, London, UK, cat. PB20.21) and dye-labeled fluorogenic reporter oligonucleotide probes of the sequences listed in table (Table 1). The PCR amplification was performed in 96-well optical reaction plates subjected to 40 cycles of 5 sec at 95 °C and 25 sec at 60 °C each. The expression of target genes was quantified by the QuantStudio™ 3 Real-Time PCR System (Applied Biosystems by Thermo Fisher Scientific Inc., Waltham, MA, USA) and analyzed following the 2^−ΔΔCT^-method [47] using TATA-binding protein (*TATAbp*) as an internal reference.

### 2.6. Immunohistochemistry

In accordance with the experimental conditions, H9c2 cells were cultured on chambered microscope glass slides (8-well chamber, removable, Ibidi GmbH, Gräfelfing, Germany, catalog #80841) in a density of 12 × 10^3^ cells per well. Cells were washed three times in PBS and then fixed in 4% paraformaldehyde (PFA) for 15 min at room temperature (RT). After being washed three more times in PBS, cells were permeabilized by applying 0.2% TritonX-100 for 15 min at RT. Permeabilized cells were washed in PBS, and the following primary antibodies diluted in Antibody Diluent (Agilent Technologies, Santa Clara, CA, USA, cat S080983-2) were added: rabbit anti-α-sarcomeric actinin (1:100, Abcam, Cambridge, UK, catalog cat ab137346) and mouse anti-Ki67 (1:100, Abcam, cat ab279653). After a 24 h incubation at 4 °C, the cells were washed three times in PBS, and secondary antibodies were added: goat–anti-rabbit Alexa Fluor 594 and goat–anti-mouse Alexa Fluor 488 (1:200 in Antibody Diluent, Life Technologies, Carlsbad, CA, USA, cat A11037/A11029). After a 1 h incubation at room temperature, cells were washed three times in PBS, and slides were mounted using 4′,6-diamidino-2-phenylindole (DAPI)-containing fluorescence-protecting mounting medium (ProLong Gold Antifade Mountant with DAPI, Thermo Fisher Science, cat P36935).

Slides were viewed blinded using a Keyence BZ 9000 compact fluorescence microscope with BZ-II Viewer software and BZ-II Analyzer software (Keyence, Osaka, Japan), and four separate sections per chamber were analyzed. Nuclei and Ki67-positive cells were counted manually by using Adobe Photoshop software 22.0.0 (Adobe Systems Software Ireland Limited, Dublin, Republic of Ireland) and the cell surface area was measured by using ImageJ (National Institute of Health, Bethesda, MD, USA). The data in the ROI of the control group were used as 100% values.

### 2.7. LDH-Assay and CCK8

As an indicator of cell death, LDH release into the culture medium was evaluated using a Cytotoxicity Detection Kit (Roche Diagnostics, Mannheim, Germany, cat 11644793001). As instructed by the manufacturer’s protocol, supernatant was taken, centrifuged at 100× *g* for 5 min, and subsequently applied to a 96-well plate in triplicate with a volume of 100 μL per well. After adding 100 μL of a reagent, including a catalyst and a dye solution, to each well, the plates were incubated for 30 min under protection from light. Measurement was conducted at 490 and 655 nm.

Cell Counting Kit-8 (CCK-8; Abcam, cat ab228554) was used to evaluate cell viability. H9c2 cardiomyocytes were seeded at 5 × 10^3^ cells/well in 96-well plates. After the various group treatments and conditions, 10 μL of CCK-8 solution was added to each well followed by 1 h incubation at 37 °C, according to the manufacturer’s instructions, and then read for optical density at 460 nm using a microplate spectrophotometer.

### 2.8. Statistical Analyses

Analysis of the data was carried out with GraphPad Prism 8.0 software (GraphPad Software, La Jolla, CA, USA). Data were analyzed using a multivariate repeated measures analysis of variance (ANOVA). Depending on which ANOVA test was used, multiple comparisons of means were carried out using Bonferroni’s post hoc test. GraphPad Prism Software was used for the generation of graphs. Data are presented as box and whisker plots, with the line representing the median while whiskers show the data variability outside the upper and lower quartiles. Differences were considered statistically significant at a *p*-value of <0.05.

## 3. Results

### 3.1. Proliferation Capacity and Hypertrophy

To determine the proliferative capacity relative to cell number and hypertrophic effects of oxygen-stressed H9c2 cardiomyocytes, we stained H9c2 cells with anti-Ki 67, anti-sarcomeric alpha actinin, and DAPI. Submitting H9c2 cells to both hyperoxia and hypoxia for 24 h had negative effects on their proliferative capacity; additionally, hyperoxic exposure resulted in an increase in cell area (hypertrophy) (Figure 1).

Compared to the normoxic control group, the ratio of Ki67-positive cells diminished by half in either case; however, the administration of DEX (1 µM) led to a modest increase in Ki67-positive cells in H9c2 cells submitted to 5% O_2_ (Figure 2A). The hypertrophic phenotype of H9c2 cells was induced by hyperoxia without a protective DEX effect (Figure 2B). Consistent with reduced mitotic activities, both H9c2 cells and NRCMs showed a significant decrease in *CycD2* gene expression under 5% O_2_ and 80% O_2_ that was restored to normoxic levels after DEX treatment (10 µM) (Figure 2A,B). Lower DEX concentrations demonstrated effects in NRCMs and H9c2 cells submitted to hypoxia (1 µM) and in hyperoxia exposed H9c2 cells (0.1 µM, 1 µM). The administration of DEX did not alter the expression of *CycD2* in cells cultivated at 21% O_2_ (Figure 2A,B). The complete data are presented in Appendix A.

### 3.2. Cell-Death-Associated Factors

#### 3.2.1. LDH Release and Cell Viability

Cardiomyocyte cells were cultured under normoxic, hypoxic, and hyperoxic conditions, and after 24 h, the release of lactate dehydrogenase (LDH) into the culture medium was measured for quantitative determination of when the plasma membrane was damaged (Figure 3A,B). In H9c2 cells, no statistically significant alterations in LDH were detected in either hypoxia or hyperoxia (Figure 3A). Interestingly, in NRCMs, submission to 80% O_2_ for 24 h led to a two-fold increase in the LDH release, while hypoxia did not have any effects (Figure 3B). The administration of DEX did not alter the LDH release in any case (Figure 3A,B).

To assess viability, H9c2 cells were incubated in 5%, 21%, or 80% oxygen for 24 h and then subjected to CCK 8 analysis (Figure 3C). Based on the detection of dehydrogenase activity in viable cells, cell viability was decreased in H9c2 cells submitted to 80% O_2_. The administration of DEX did not produce significant alterations in cell viability. Hypoxia did not influence cell viability either (Figure 3C). The complete data are presented in Appendix A.

#### 3.2.2. Apoptosis and Autophagy

The extrinsic apoptotic pathway is mainly mediated via the dimerization and activation of initiator caspase 8 (Casp8) secondary to extracellular apoptotic stimuli. It subsequently activates executioner caspases, such as caspase 3 (Casp3), to initiate cellular degradation [48]. Under hyperoxia and hypoxia, *Casp3* was upregulated in both H9c2 cells (Figure 4A) and NRCMs (Figure 4B). DEX reduced the expression of *Casp3* at all concentrations and under either O_2_ condition, effectively reaching values located below the normoxic control group (e.g., in NRCMs under 5% O_2_ with 0.1 µM DEX). *Casp8* was upregulated in cells submitted to hyperoxia regardless of the cell type (Figure 4C,D); however, a statistically significant upregulation under hypoxia was only observed in NRCMs (Figure 4D). DEX induced a potent downregulation of the *Casp8* expression at all DEX concentrations (0.1 µM, 1 µM, 10 µM) in H9c2 cells under both 5% O_2_ and 80% O_2_ and in NRCMs submitted to hypoxia (0.1 µM, 1 µM, 10 µM) or hyperoxia (0.1 µM). The expression of *Casp3* and *Casp8* was not changed by DEX in cells cultivated under 21% O_2_ (Figure 4C,D).

The apoptosis-inducing factor (AIF) is known as an effector of intrinsic-caspase-independent apoptotic cell death [49]. In H9c2 cells, hypoxia and hyperoxia led to an upregulation of *AIF* expression (Figure 4E). DEX restored normoxic levels at all concentrations. Interestingly, both hyperoxia and hypoxia exerted contrary effects on NRCMs (Figure 4D). Here, we observed a marked downregulation of *AIF* in either case. In NRCMs, DEX (1 µM and 10 µM) induced a modest increase in *AIF* expression under hypoxia and showed no effects under hyperoxia. Autophagy is of pivotal importance to cardiomyocytes for the maintenance of cellular homeostasis and in the generation of cardiocirculatory diseases. Autophagy-related 5 (Atg5) and autophagy-related 12 (Atg12) play a crucial role in the formation of the autophagosome [50]. We found that in H9c2 cells, the expression of *Atg5* and *Atg12* was upregulated in cells cultivated at both 5% O_2_ and 80% O_2_ for 24 h (Figure 5A,C). DEX induced a decrease in the expression of both transcripts, reaching control levels at all DEX and oxygen concentrations except for *Atg5* in hypoxia-exposed H9c2 cells treated with 1 µM DEX (Figure 5A). Intriguingly, diametrically opposed impacts on NRCMs were determined (Figure 5B,D). Both hypoxia and hyperoxia reduced the expression of *Atg5* and *Atg12*. No statistically significant alterations in the expression of *Atg5* were found in NRCMs after the administration of DEX (Figure 5B). The downregulation of *Atg12* in NRCMs under 80% O_2_ was compensated by DEX at a concentration of 1 µM (Figure 5D). DEX did not cause any alterations in the expression of *Atg5* or *Atg12* in cells submitted to normoxia. The complete data are presented in Appendix A.

### 3.3. Oxidative Stress Response

The protein GCLC acts as the catalytic subunit of the glutamate cysteine ligase (GCL), an enzyme that catalyzes the first and rate-limiting step of glutathione synthesis. It is therefore involved in cellular antioxidative mechanisms [51]. Hyperoxia led to a three-fold increase in the expression of *GCLC* in NRCMs (Figure 6B) and to a marked upregulation in H9c2 cells (Figure 6A). The administration of DEX induced a decrease in *GCLC* expression at all concentrations in cells submitted to hyperoxia. A downregulation of *GCLC* under hypoxic conditions was only detected in NRCMs (Figure 6B). At atmospheric oxygen levels, DEX did not alter the expression of *GCLC* in H9c2 cells (Figure 6A), whereas in NRCMs, the administration of 1 µM DEX led to a modest downregulation of *GCLC* expression (Figure 6B).

Oxidative stress puts cellular homeostasis at risk. Nuclear factor-erythroid 2-related factor 2 (Nrf2) is a transcription factor that mediates a plethora of cytoprotective antioxidative mechanisms [52]. Hyperoxia induced a marked increase in *Nrf2* expression in H9c2 cells (Figure 6C), reaching peak values of approximately 200% compared to the control group. DEX restored normoxic levels of *Nrf2* at all concentrations (0.1 µM, 1 µM, 10 µM). In H9c2 cells submitted to normoxia, all DEX concentrations led to a decrease in the expression of *Nrf2* (Figure 6C). The upregulation of *Nrf2* in cells cultivated under 80% O_2_ was less intense in NRCMs, and the administration of DEX had no effect on the expression of *Nrf2* detected (Figure 6D).

The cellular adaptation to conditions of low oxygen is orchestrated by hypoxia-inducible factor 1α (Hif1α) [53]. As expected, we detected an upregulation of *Hif1α* in NRCMs and H9c2 cells submitted to hypoxia (Figure 6E,F). DEX led to a decrease, reaching levels measured in the control group at all concentrations (0.1 µM, 1 µM, 10 µM) regardless of the cell type. The expression of *Hif1α* was not affected by hyperoxia. In H9c2 cells, DEX (1 µM) induced a statistically significant, but modest decrease in the expression of *Hif1α* (Figure 6E). The complete data are presented in Appendix A.

### 3.4. Cardiomyocyte Tissue Structure

The cardiomyocyte-specific marker cardiac muscle troponin T (Tnnt2) is a sensitive marker of cardiac injury and plays an important role in the regulation of cardiac muscle contraction [54]. The expression of *Tnnt2* was upregulated in H9c2 cells when submitted to 80% O_2_ (Figure 7A). DEX showed lowering effects at all doses. In contrast, in NRCMs, cells cultivated under hyperoxic conditions showed a decrease in the expression of *Tnnt2* by half compared to the control group (Figure 7B). Hypoxia had no effects on *Tnnt2* of H9c2 cells and NRCMs. In both cases, the administration of DEX did not lead to any alterations (Figure 7A,B).

Tissue inhibitor of metalloproteinases (TIMPs) maintains the homeostatic balance of myocardial ECM by inhibiting activated matrix metalloproteinases (MMPs) that contribute to myocardial fibrosis through different mechanisms [55]. When cultivated under 80% O_2_, both H9c2 cells and NRCMs showed an upregulation of *Timp1-* and *Timp2*-expression (Figure 7C–F). In all cases, administration of DEX effectively lowered the increased expression of *Timp1* and *Timp2* at 0.1 µM. Higher doses (1 µM and 10 µM) showed statistically significant effects on every group except for *Timp2* in NRCMs. Hypoxia did not have any effect on the expression of *Timp1*, regardless of the cell type (Figure 7C,D). In hypoxia-exposed H9c2 cells, no alterations of the expression of *Timp2* were detected (Figure 7E). In NRCMs, however, *Timp2* was expressed to a lesser extent in NRCMs submitted to hypoxia. Under these circumstances, DEX (10 µM) led to an increase in *Timp2* expression (Figure 7F). The complete data are presented in Appendix A.

### 3.5. Hippo Pathway

The Hippo signaling pathway is known to play a critical role in the cardiomyocyte cell cycle [56]. Briefly, a kinase cascade containing Mst1 and Lats2 leads to the phosphorylation of Yap, preventing its nuclear translocation. Consequently, Yap can neither associate with its cofactors, such as Tead1, nor bind to the DNA to act as a transcription factor with positive effects on cell cycle progression [56].

We found that *Cul7*, a ubiquitin ligase that promotes the degradation of Mst1, is expressed at lower levels under 5% O_2_ and 80% O_2_ compared to the control group in both H9c2 cells and NRCMs (Figure 8A,B). Treatment with DEX (1 µM and/or 10 µM) increased the expression up to the normoxic level in H9c2 cells (80% O_2_ and 5% O_2_, Figure 8A), whereas a significant increase with DEX (1 µM and 10 µM) in NRCMs could be detected in cells cultivated under 5% O_2_. DEX did not cause any alterations in cells submitted to 21% O_2_.

In H9c2 cells, both hyperoxia and hypoxia led to an induction of *Lats2* expression. Treatment with DEX at low doses (0.1 µM) induced a decrease in either case. Higher doses (1 µM and 10 µM) showed statistically significant effects in cells submitted to hyperoxia for 24 h (Figure 8C). In NRCMs, the expression of *Lats2* was increased after incubating the cells at 80% O_2_, whereas, despite a recognizable tendency towards higher expression of *Lats2*, no statistical significance could be detected under hypoxic conditions. After the application of DEX, *Lats2* levels decreased at all DEX concentrations for the hypoxic group and at 0.1 µM and 1 µM for the hyperoxic group (Figure 8D). DEX did not change the expression of *Lats2* at 80% O_2_.

Hyperoxia downregulated the expression of *Tead1* in NRCMs and H9c2 cells (Figure 8E,F), whereas a statistically significant decrease under hypoxia could be detected in H9c2 cells (Figure 8E). In H9c2 cells, DEX enhanced the expression of *Tead1* under hypoxic (10 µM) and hyperoxic (1 µM, 10 µM) conditions (Figure 8E). In NRCMs, this effect was discovered after the administration of 0.1 µM DEX at 80% O_2_.

Interestingly, the expression of *YAP1* was markedly upregulated in both H9c2 cells and NRCMs submitted to hyperoxia for 24 h (Figure 8G,H). At all concentrations, DEX led to a decrease in *YAP1* and reached normoxic control levels. Hypoxia did not alter the expression of *YAP1*. The complete data are presented in Appendix A.

## 4. Discussion

In this study, we demonstrated that compared with ambient air, lower oxygen concentration (hypoxia or physioxia) and higher oxygen tension (hyperoxia) decreased the proliferative capacity and associated transcripts of immortalized and primary rat cardiomyocytes, and hyperoxia-induced cellular hypertrophy. Apoptosis-related transcripts were upregulated under both hypoxic and hyperoxic conditions. Autophagy-related mediators were induced for immortalized H9c2 cells, whereas these were downregulated for primary cardiomyocytes (NRCM). This was consistent with the induction of oxidative-stress-response-related mediators in response to hypoxia as well as hyperoxia. Cardiac stress and tissue remodulation factors appeared to be more influenced by hyperoxia. DEX was able to counteract these oxygen-modulated effects at the transcript level in both immortalized and primary rat cardiomyocytes. Beside redox-sensitive signaling mediators, the Hippo signaling pathway affected the cardio-protective effects.

Oxidative stress is an imbalance between pro-oxidant trigger factors and corresponding counter-regulatory mechanisms. Free radicals and ROS are formed under physiological conditions during oxidative metabolism, but in concentrations that are not toxic to cells. Insufficient antioxidant capacities of cells or tissues cause an imbalance, so that ROS accumulate. Increased ROS causes damage to macromolecules, as a result of which cell death, proliferation, or cell differentiation factors and mediators are regulated and involved in the pathogenesis of disease in the premature setting [22,23,57]. Abnormal production of free radicals causes alterations in molecular signaling pathways that underlie the development of cardiovascular diseases [58,59]. In the postnatal situation, a rapid adaptation of the heart from lower in utero oxygen to a higher ex utero oxygen supply occurs and causes a proliferation arrest of cardiomyocytes. The change in oxygen partial pressure with increased ROS is considered causative for myocardial cell cycle arrest [25]. Hypoxia is essential for the developing heart, whereas decreased oxygenation in the adult heart leads to cell death, cardiomyocyte hypertrophy, fibrosis, and impaired cardiac function. Hypoxia critically modulates the expression of genes responsible for cell death, metabolism, proliferation, and differentiation. An important player in the regulation of these processes is the transcription factor Hif1α [26]. Another redox-sensitive transcription factor is Nrf2, which is involved in regulating the expression of genes for antioxidants and detoxification [60].

In parallel with cardiomyocytes isolated from fetal hearts, embryonic rat ventricular myocyte H9c2 cells were used to study proliferation and regulation of various transcripts [61]. Our in vitro study clearly showed the impairment of proliferation by low and high oxygen concentrations, as well as the induction of a hypertrophic phenotype after hyperoxic exposure of the immortalized cells. The direct hypoxia/hyperoxia proliferation inhibitory effect on H9c2 myocytes was confirmed by immunofluorescence staining of the nuclear protein Ki-67, which is expressed in all cell cycle phases except G0 [62]. Quantitative analysis of α-sarcomeric actinin and Ki67-positive cells suggested that hypoxia and hyperoxia reduced proliferative capacity, although complete cell cycle arrest could not be inferred. An induction of expression of relevant transcription factors was demonstrated for Nrf2 for hyperoxia and for Hif1α for hypoxia. This is consistent with findings from an in vitro study of neonatal cardiomyocytes, in which 1% oxygen resulted in the induction of Hif1α but inhibited mitotic activities and induced cell death [63]. Furthermore, already maternal hypoxia (10.5% O_2_) in pregnant mice caused cell cycle arrest in fetal cardiomyocytes [64].

In line with this, 24 h of hypoxia conditioned an increase in Casp3 and Casp8 gene expression for both cell lines and primary cells, which is significant for caspase-dependent apoptosis. Hypoxia serves the Hif1α signaling pathway for progression of proliferation as well as regulation of cardiomyocyte apoptosis, according to these results. In the complex orchestration of the regulation of cellular processes in cardiomyocytes, other mediators must be involved. The apoptosis-related caspase-independent factor AIF and the autophagy-involved members of the ATG family, Atg5 and Atg12, were induced by hypoxia in immortalized H9c2 cells, suggesting that differential pathways may trigger cell death. Considering the results obtained for primary cardiomyocytes, hypoxia induced the caspase-dependent apoptosis factors Casp3 and Casp8 as well but reduced the expression of AIF and the autophagy mediators Atg5/12. Numerous studies document differential types of cell death, such as apoptosis, autophagy, and ferroptosis, in myocardial ischemia [65,66,67]. The underlying mechanisms for myocyte cell death are not completely understood.

Lysosome-mediated autophagy has certainly been linked to cardiac disease, with both a degenerative and protective effect seeming possible [68]. Impaired progression of autophagy appears to be a factor in myocardial injury [68,69] and is orchestrated by an overlapping of a variety of signaling pathways (for an overview, see [70]). Hif1α also appears to be autophagy-regulatory via the Hif1α/BNIP3 pathway. Shown in H9c2 cells under OGD/R injury as well as in CoCl_2_-induced injury to mimic hypoxia [71], both Hif1α and autophagy-related mediators are induced [72]. Compared to previous studies, DEX demonstrated a protective effect on cardiomyocytes by inhibiting ROS-induced apoptosis [41], an antioxidant effect on the developing rodent brain [73], as well as known neuroprotective mechanisms in general [74,75]. Li et al. [76] demonstrated the protection of H9c2 cells against CoCl_2_-induced damage using the antioxidant naringin, a natural bioflavonoid, by enhancing autophagic flux via activation of the Hif1α/BNIP3 pathway. In our data, transcripts for Hif1α as well as Atg5/12 are decreased by DEX under hypoxia. Hif1α induction also occurred by hypoxia in primary NRCMs; however, in this primary cell type DEX tended to induce the expression of autophagicAtg5/12.

Secondly, Hifs regulate other cellular processes, such as energy substrate metabolism. In this context, Hif1α, triggered by hyperoxia, induces the expression of lactate dehydrogenase (LDH) [77]. We detected an LDH increase only for primary NRCMs under hyperoxia, but without Hif1α regulation. It should be noted here that the primary NRCMs had to withstand very high-stress levels under hyperoxia, recognizable by a high number of intracellular vesicles. A tendency may be discussed for LDH induction with increased Hif1α for the H9c2 cells, which might support this statement. In a low oxygen environment, in vivo associated with the in utero situation of the fetal maturing heart, and Hif1α is again a game changer when it comes to the transition of oxygen concentration. Hif1α controlling cardiac energy metabolism may have far-reaching consequences for the newborn heart in a premature situation due to the multiple effects of the targeted genes. [78]. Postnatally, the proliferation rate of cardiomyocytes is drastically reduced as hypoxia-sensitive transcription factors decrease; without growth stimuli, cardiomyocytes hypertrophy. Compared with physiological hypertrophy, progressive pathophysiological hypertrophy is associated with cellular changes, including marked changes in gene expression [78].

Nrf2 is another important transcription factor involved in redox homeostasis and oxidative stress response. The Nrf2 signaling pathway is equally implicated in the interactions between autophagy and cardiovascular disease. By inhibiting oxidative stress responses, Nrf2 signaling may be beneficial to the regulation of stability, with clear evidence that stimulation of Nrf2 signaling pathways may prove cytoprotective [79,80]. Our in vitro analyses revealed an induction of Nrf2 transcription under hyperoxic insult for both the cell line and primary cardiomyocytes. This could be downregulated by DEX in the H9c2 myocytes and displayed a decreasing trend in the primary NRCMs. The transcription factor Nrf2 not only regulates the oxidative stress response but is also an anti-inflammatory protein and exhibits pro-inflammatory properties under different physiological circumstances. Upregulated Nrf2 often shows up in the early damage phase and can enhance myocyte injury and apoptosis [81]. Whether Nrf2 up-regulation is the initial event or represents a secondary response is difficult to assess. The hypothesis of constitutively activated Nrf2 as well as Nrf2 induced under oxidative stress supports the contention that Nrf2 represents an important regulator by activating nearby antioxidant genes, such as *heme oxygenase * (HO1) and *superoxide dismutase* (SOD) [79].

In this context, it may also be mentioned that Nrf2 expression is upregulated in the early stages of myocardial remodeling and is connected to apoptosis, fibrosis, myocardial hypertrophy, and cardiac dysfunction [82]. What we found is that under hyperoxic conditions, Timp1 and Timp2 were generated at increased transcript levels, both for H9c2-myocytes and primary cardiomyocytes. Remodeling of cardiac tissue and resulting dysfunction are causative for hypertrophic progression of cardiac tissue. It is quite promising that abnormal autophagy also appears to be involved. If hypertrophy is mimicked in primary, neonatal mouse cardiomyocytes and autophagy is subsequently reduced by Atg5-siRNA, the connection can be reconstructed via impaired contractility capacity [83]. As mentioned previously, maternal hypoxia inhibits cardiomyocyte proliferation in fetal and neonatal rat pups [64]. Furthermore, hypoxia has been shown to directly inhibit proliferative capacity via initiation of the gene expression of TIMPs. This was shown in our in vitro study for exposure to high oxygen concentrations; i.e., upregulated Timps1/2 is in line with decreased proliferation capacity and downregulation of CycD2 [63,64].

The cardiomyocyte marker Tnnt2, relevant to cardiomyocyte functions, such as contractility, was not modulated by hypoxia in our in vitro experiments, although it was differentially regulated under hyperoxia. H9c2 myocytes upregulated Tnnt2, whereas in primary NRCM, Tnnt2 transcription was inhibited. DEX exhibited counter-regulatory effects only for the immortalized cells. Cardiomyocytes deleted for the heterozygous and homozygous Tnnt2 gene exhibited decreased contractility compared with wild-type cardiomyocytes [84]. Cardiac diseases, such as dilated cardiomyopathy or hypertrophic cardiomyopathy, are progressive, and it is suspected that genes involved in the earliest disease development and divergence may be involved.

Hyperoxic insult as an oxidative-stress triggering factor has not been adequately studied for effects on cellular cardiac impairments. Affected cardiac functions can be demonstrated in experimental studies in mice [85]. It has been clearly shown that neonatal hyperoxia also resulted in cardiac dysfunction in adult mice [86], similar to clinical studies of premature infants with cardiac dysfunction in adults [14]. Hyperoxia-modulated redox homeostasis and its downstream signaling pathways, such as Nrf2 signaling pathways [79,80], as previously elaborated, may be causative [87,88]. Hyperoxia affects postnatal proliferation of cardiomyocytes similarly to hypoxia. Neonatal mice exposed to 100% O_2_ during the first four days of life inhibited cardiomyocyte proliferation, resulting in hypertrophy [89]. Adult-human-transplant-derived cardiomyocytes responded to 48 h of hyperoxia at 95% O_2_ with increased cell death as well as increased LDH levels [90], similar to our primary NRCMs.

The effects of oxidative stress, in terms of hyperoxia as well as hypoxia, condition individual interactions with redox-sensitive signaling pathways and the complex interconnectedness among these pathways [21]. Cardiomyocytes grow during embryonic development and are regulated by different signaling pathways. Nevertheless, it is worth looking at a signaling pathway that plays an important function during cardiac development, such as the Hippo/Yap signaling pathway (for review see [91,92,93]). The Hippo–YAP pathway plays a critical role in morphogenesis by regulating cardiomyocyte proliferation and differentiation [91,92,94].

The Hippo signaling pathway is affected by ROS-induced oxidative stress. This changes the transcription of mediators, such as YAP1, which is the major Hippo downstream target. In primary mouse cardiomyocytes, overexpression of YAP caused antiapoptotic effects on H_2_O_2_-induced oxidative stress [95]. In cardiomyocytes themselves, YAP1 may act as a transcriptional co-activator, contributing to the upregulation of antioxidant genes [96]. YAP1 transcription was induced to a considerable extent only by 80% oxygen in our in vitro experiments, but was also regularly reduced with DEX, whereas no changes were observed under hypoxia. Mammalian sterile 20-like kinase 1 (Mst1) and Lats2, players in the Hippo signaling pathway, regulate phosphorylation of YAP in the heart and its nuclear translocation with subsequent transactivation of cell cycle genes [97]. Mst1 has been proven to influence myocardial autophagy and strongly suggests the influence of the Hippo signaling pathway on the regulatory processes of autophagy and apoptosis [98,99]. Concomitant with the upregulation of YAP1, induction of Lats2 expression occurred in both H9c2 cells and NRCMs, but here also increased expression under low oxygen (5%). Cul7 appears to adopt pro-survival and proliferation-promoting properties in the heart [100,101]; Cul7 depletion caused hypertrophy in cardiomyocytes [102]. Moreover, Cul7 promotes the Mst1 degradation in cardiomyocytes [97]. Cul7 expression was in line with decreased cardiomyocyte proliferation in H9c2 and primary cells. DEX was able to enhance downregulated Cul7 transcripts under hypoxia for H9c2 and NRCM, as well as for H9c2 under hyperoxia. After translocation of YAP1 to the nucleus, it acts with various transcription factors, such as TEAD1, to regulate the targets involved in cardiac proliferation, survival, and differentiation [91,92,94]. The expression of YAP1 is positively correlated with the change in cardiomyocyte proliferation in response to hypoxic conditions and seems to be related to Tnnt2 expression [103]. TEAD1 mRNA expression showed a reduced level for both oxygen concentrations and could be counter-regulated by DEX. Tnnt2 acts differently and is shown to be affected only by hyperoxia. H9c2 cells induced Tnnt2, whereas primary cardiomyocytes decreased in Tnnt2 expression. The sensitive relationships between Hippo signaling and cardiomyocytes were evident with YAP1 inactivation. This decreased cardiomyocyte proliferation in vivo and in vitro, whereas fetal activation of YAP1 promoted it, although physiological hypertrophy was not affected [104].

We were able to demonstrate a great vulnerability of immortalized as well as primary fetal cardiomyocytes to oxidative stress triggered by both low and high oxygen concentrations compared with normoxic controls. Direct comparison to the influence of oxygen concentrations is limited in some aspects for the cells used. In several experimental approaches, we have illustrated that the immortalized rat H9c2 cardiac cell lines differ markedly from the corresponding neonatal primary cardiac cells in some aspects and resemble each other in other aspects [105,106]. Thus, both comparable effects of oxidative stress on apoptosis-associated mediators were identified, but concomitantly, the effects of aberrant oxygen concentrations on autophagy-associated mediators differed. This was well correlated with effects on Hippo pathway transcripts and impaired proliferation.

Exciting in this context is the influence of DEX, which could congruently counteract the oxidative stress responses almost everywhere. The results were very impressive for the transcript expression of the cell cycle regulator cyclin D2 and cell-death-associated mediators with direct reference to the Hippo pathway. A possible link with accentuated evidence for autophagy could be discussed in light of recent findings on mechanisms for DEX (reviewed in [93]) always in terms of the antioxidant properties of DEX, per se. In addition, some other antioxidants, such as N-acetylcysteine, were able to promote proliferation of neonatal cardiomyocytes in vitro and in vivo by activating the YAP signaling pathway. The dependence of high oxidative metabolism for proliferation of neonatal cardiomyocytes may account for the vulnerability to varying oxygen concentrations [93,107]. It should also be mentioned that cardiometabolic disease, with a higher prevalence in preterm infants, is a multifactorial process. Factors leading to overproduction of free radicals and thus oxidative stress include total parenteral nutrition, blood transfusions, as well as painful procedures (reviewed by [108]).

Limiting for our study in immortalized as well as primary cardiomyocytes are the oxygen concentrations because they cannot fully mimic the individual tissue pO_2_, named physioxia, since, on the one hand, concrete values for preterm infants are not available and, on the other hand, the ambient conditions of physioxia for standard isolation from rat hearts as well as the standard cell line could not be realized. Moreover, no confirmatory proteomic analyses were performed in consideration of the reduction in animal numbers according to the 3R principle, as gene expression profiling has sufficient potential to reveal the effects of exogenous factors by analyzing secondary responses to conditional regulatory events. Known cellular process- or pathway-dependent biomarkers can fundamentally describe the cellular effects that subsequently drive degenerative or protective cell fates. Importantly, we experimented with undifferentiated H9c2 cells, which may differ markedly from primary cardiac cells in morphology and may also differ in gene expression patterns. Even differentiation of H9c2 cells could not have abolished this difference. Because there is an essential need for new insights into molecular mechanisms for multifaceted cardio-associated diseases, cell lines can be used to clarify fundamental aspects. Of course, this is also done to identify alternative methods in line with the 3Rs principle. To be able to use valid data for in vitro studies and to reveal differences between tissues, primary cells and target lines that are widely used in experimental research, it is of high importance to study detailed knowledge of the strengths of cardiac cell lines. There is a significant need for experimental models to understand why high oxygen exposure at birth is a risk factor for adult cardiovascular disease.

## 5. Conclusions

Cardiovascular impairment of preterm infants in adulthood may be caused by changes in cellular functionality during prematurity. An early disturbance of cardiomyocyte proliferation in favor of hypertrophic cells and a subsequent reduction of the cell pool due to an early hypoxic–hyperoxic transition would be a debatable aspect. The reduction in oxidative stress by antioxidant drugs as well as their protective effects on cellular degeneration could be a therapeutic target. The balance of the necessary or detrimental effects of oxygen is the target of diverse experimental approaches in neonatology research. The formation of ROS is not avoidable in premature birth, but it does have long-term effects on cerebral, pulmonary, and cardiovascular structural and functional organ development.

Our experimental procedures with different oxygen concentrations support this controversial debate on the role of oxygen in the effects on the preterm heart. In fact, high oxygen concentration is not found to be beneficial for cardiac development. Interestingly, in this study, we present results showing that cell cycle regulators of neonatal cardiomyocytes in vitro can be increased by pretreatment with dexmedetomidine under low and high oxygen, but the cellular “inhibitory” role of high (80% O_2_) as well as low oxygen (5% O_2_) concentration on cardiomyocyte proliferation was not clearly comprehensible; however, what is considered the normal oxygen concentration is relative, and the discussion could also start from the possibly normal 5% oxygen relative to moderate (21% O_2_) and absolute (80% O_2_) hyperoxia.

Therefore, it is crucial to determine the optimal oxygen concentration for each clinical situation to identify oxidative-stress-related late effects well and to establish new therapeutic strategies.

## Figures and Tables

**Figure 1 antioxidants-12-01206-f001:**
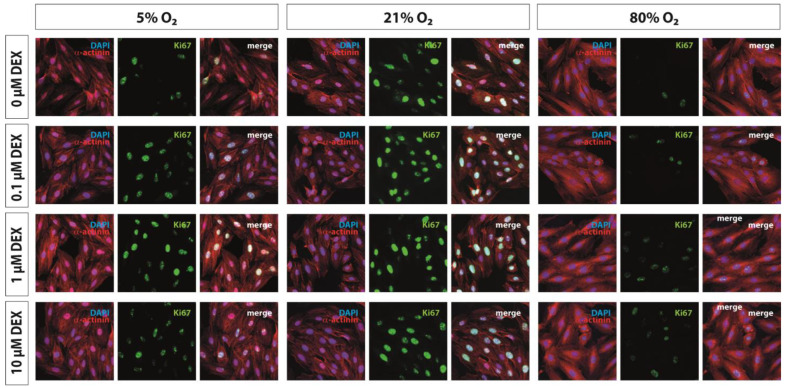
Reduced proliferation capacity caused by hypoxia and hyperoxia, as well as induction of hypertrophy by hyperoxia of H9c2 cardiomyocytes. Dexmedetomidine (DEX) was found to enhance hypoxia-induced decreased proliferation. Representative images of co-labeled H9c2 cells with Ki67 (green), α-sarcomeric actinin (α-actinin, red) and DAPI (blue) exposed to normoxia (21% O_2_), hypoxia (5% O_2_) or hyperoxia (80% O_2_) compared to H9c2 cells treated with dexmedetomidine (DEX; 0.1 µM, 1 µM, 10 µM). Analyses were conducted after 24 h oxygen exposure. Magnification of 20×.

**Figure 2 antioxidants-12-01206-f002:**
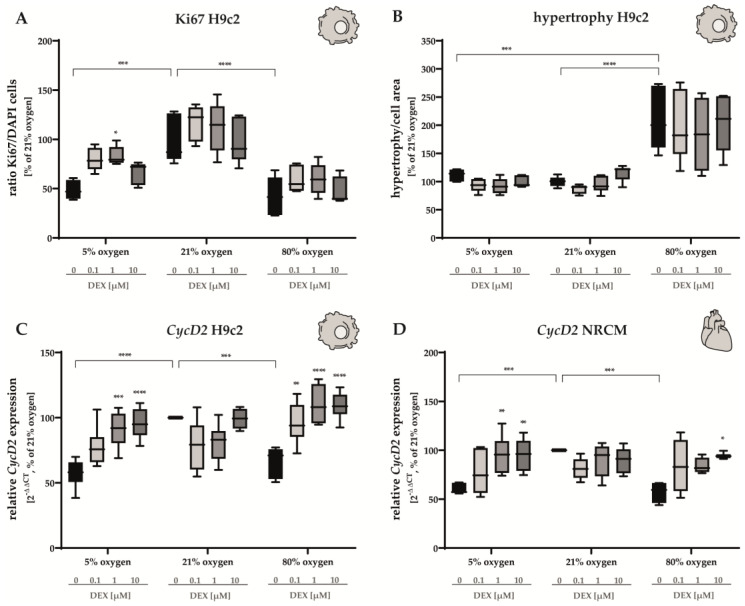
Immunocytochemical quantification of (**A**) counts of co-labeled Ki67/DAPI-positive H9c2 cells and (**B**) hypertrophy of H9c2 cells as well as expression of (**C**) *CycD2* of H9c2 cells and (**D**) *CycD2* of NRCMs were performed for normoxia (21% O_2_), hypoxia (5% O_2_) and hyperoxia (80% O_2_). Quantitative analysis was conducted for the DEX-treated cells with the concentrations 0.1 µM (light gray bars), 1 µM (gray bars) and 10 µM (dark gray bars) in comparison to the untreated control group (black bars). Data are normalized to the level of cells exposed to normoxia (21% O_2_) at each time point (control 100%, black bars). n = 6 individual experiments/group. * *p* < 0.05, ** *p* < 0.01, *** *p* < 0.001, **** *p* < 0.0001 (ANOVA, Bonferroni’s post hoc test).

**Figure 3 antioxidants-12-01206-f003:**
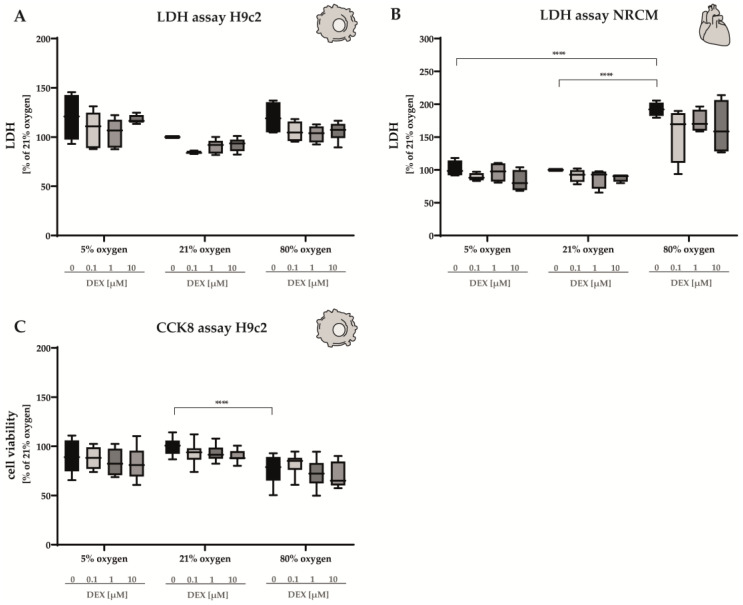
LDH release of (**A**) H9c2 cells and (**B**) NRCMs as well as (**C**) cell viability of H9c2 cells were performed for normoxia (21% O_2_), hypoxia (5% O_2_) and hyperoxia (80% O_2_). Quantitative analysis was conducted for the DEX-treated cells with the concentrations 0.1 µM (light gray bars), 1 µM (gray bars) and 10 µM (dark gray bars) in comparison to the untreated control group (black bars). Data are normalized to the level of cells exposed to normoxia (21% O_2_) at each time point (control 100%, black bars). n = 6 individual experiments/group., **** *p* < 0.0001 (ANOVA, Bonferroni’s post hoc test).

**Figure 4 antioxidants-12-01206-f004:**
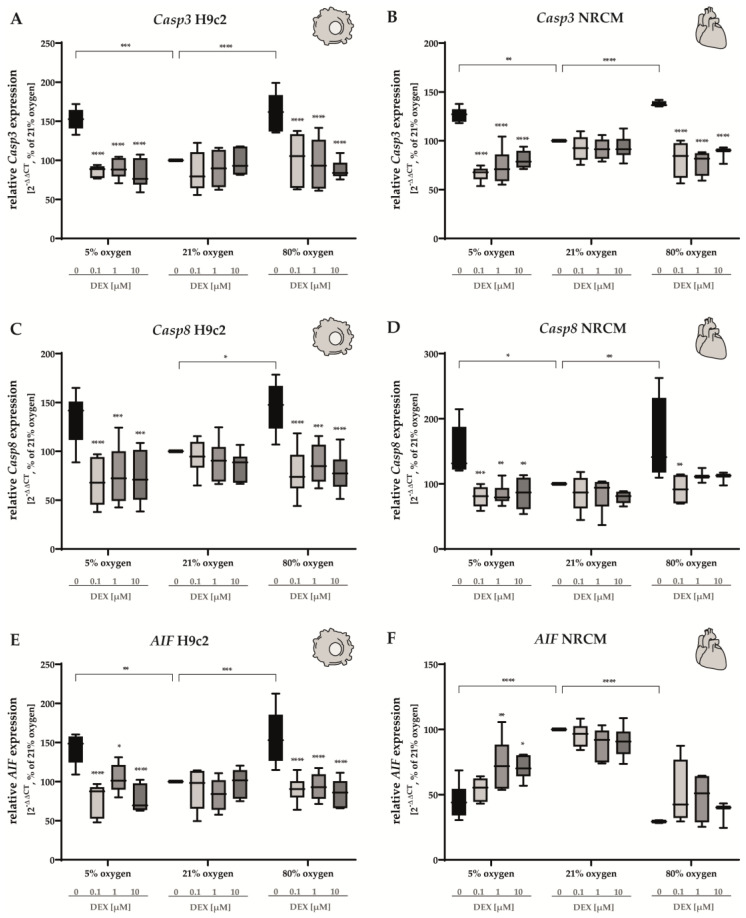
Expression of (**A**,**B**) *Casp3*, (**C**,**D**) *Casp8*, and (**E**,**F**) *AIF* of H9c2 cells (left column) and of NRCMs (right column) were performed for normoxia (21% O_2_), hypoxia (5% O_2_) and hyperoxia (80% O_2_). Quantitative analysis was conducted for the DEX-treated cells with the concentrations 0.1 µM (light gray bars), 1 µM (gray bars) and 10 µM (dark gray bars) in comparison to the untreated control group (black bars). Data are normalized to the level of cells exposed to normoxia (21% O_2_) at each time point (control 100%, black bars). n = 6 individual experiments/group. * *p* < 0.05, ** *p* < 0.01, *** *p* < 0.001, **** *p* < 0.0001 (ANOVA, Bonferroni’s post hoc test).

**Figure 5 antioxidants-12-01206-f005:**
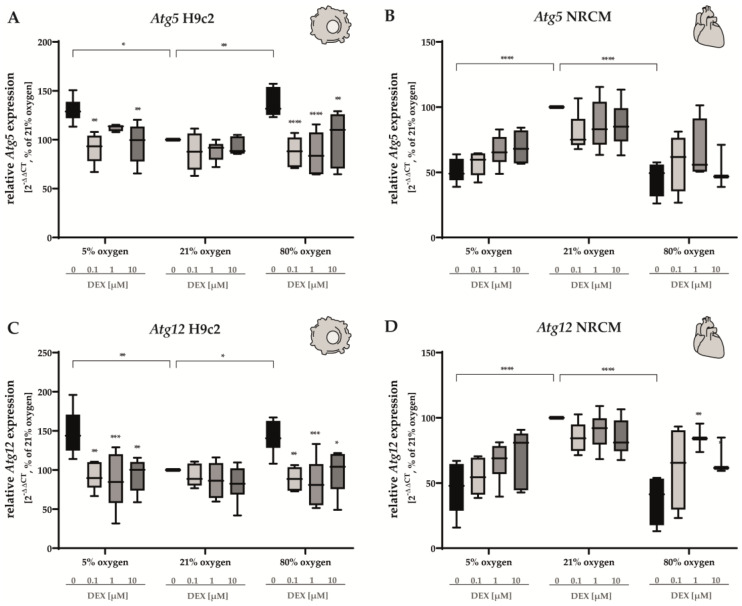
Expression of (**A**,**B**) *Atg5* and (**C**,**D**) *Atg12* of H9c2 cells (left column) and of NRCMs (right column) were performed for normoxia (21% O_2_), hypoxia (5% O_2_) and hyperoxia (80% O_2_). Quantitative analysis was conducted for the DEX-treated cells with the concentrations 0.1 µM (light gray bars), 1 µM (gray bars) and 10 µM (dark gray bars) in comparison to the untreated control group (black bars). Data are normalized to the level of cells exposed to normoxia (21% O_2_) at each time point (control 100%, black bars). n = 6 individual experiments/group. * *p* < 0.05, ** *p* < 0.01, *** *p* < 0.001, **** *p* < 0.0001 (ANOVA, Bonferroni’s post hoc test).

**Figure 6 antioxidants-12-01206-f006:**
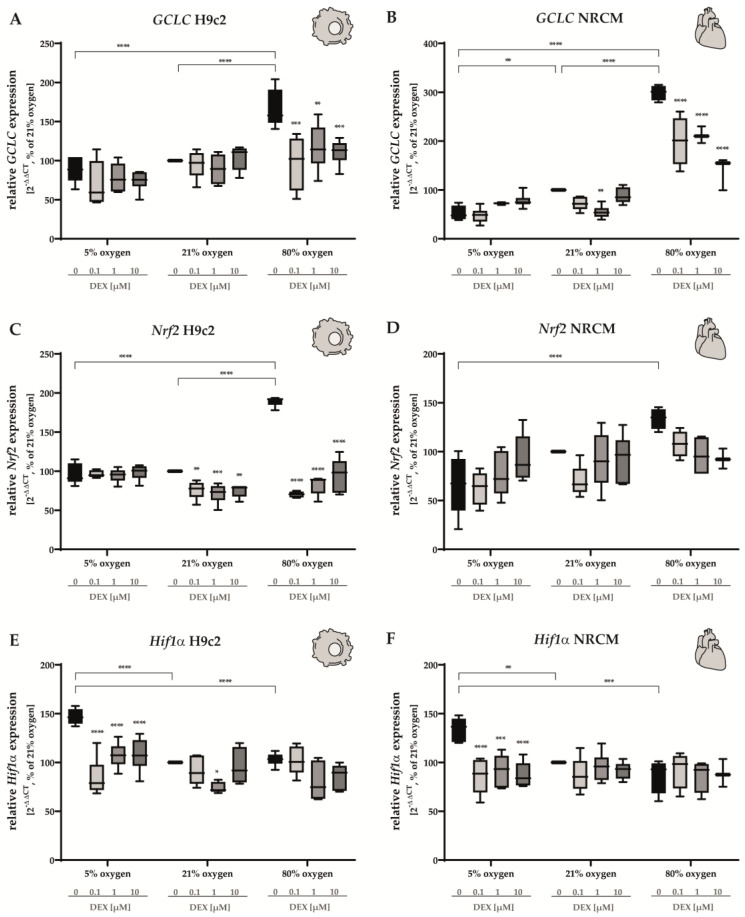
Expression of (**A**,**B**) *GCLC*, (**C**,**D**) *Nrf2*, and (**E**,**F**) *Hif1α* of H9c2 cells (left column) and of NRCMs (right column) was determined for normoxia (21% O_2_), hypoxia (5% O_2_) or hyperoxia (80% O_2_). Quantitative analysis was conducted for the DEX-treated cells with the concentrations 0.1 µM (light gray bars), 1 µM (gray bars) and 10 µM (dark gray bars) in comparison to the untreated control group (black bars). Data are normalized to the level of cells exposed to normoxia (21% O_2_) at each time point (control 100%, black bars). n = 6 individual experiments/group. * *p* < 0.05, ** *p* < 0.01, *** *p* < 0.001, **** *p* < 0.0001 (ANOVA, Bonferroni’s post hoc test).

**Figure 7 antioxidants-12-01206-f007:**
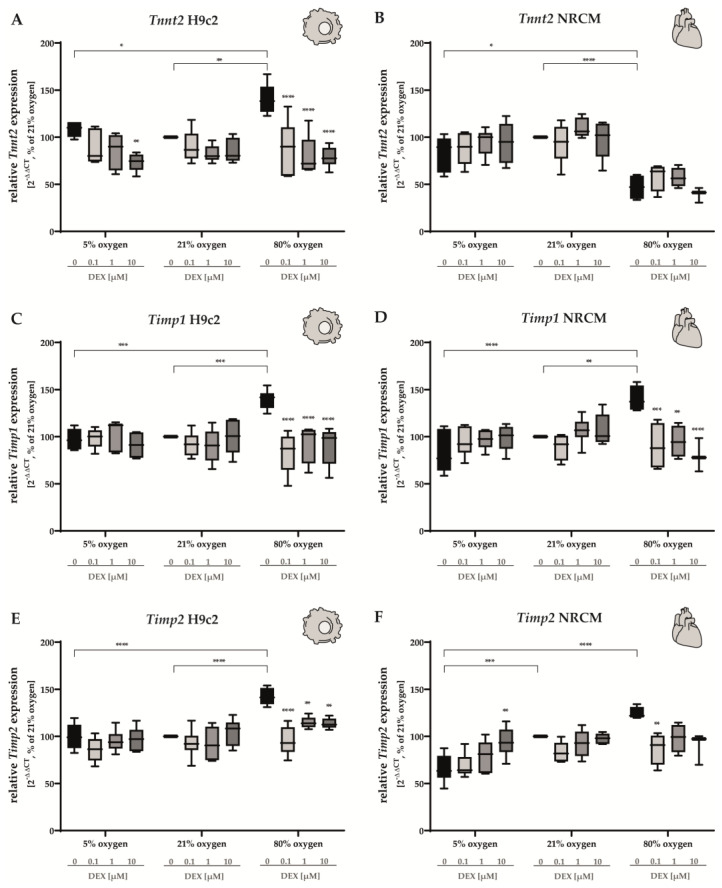
Expression of (**A**,**B**) *Tnnt2*, (**C**,**D**) *Timp1*, and (**E**,**F**) *Timp2* of H9c2 cells (left column) and of NRCMs (right column) was determined for normoxia (21% O_2_), hypoxia (5% O_2_) and hyperoxia (80% O_2_). Quantification was conducted for the DEX-treated cells with the concentrations 0.1 µM (light gray bars), 1 µM (gray bars) and 10 µM (dark gray bars) in comparison to the untreated control group (black bars). Data are normalized to the level of cells exposed to normoxia (21% O_2_) at each time point (control 100%, black bars). n = 6 individual experiments/group. * *p* < 0.05, ** *p* < 0.01, *** *p* < 0.001, **** *p* < 0.0001 (ANOVA, Bonferroni’s post hoc test).

**Figure 8 antioxidants-12-01206-f008:**
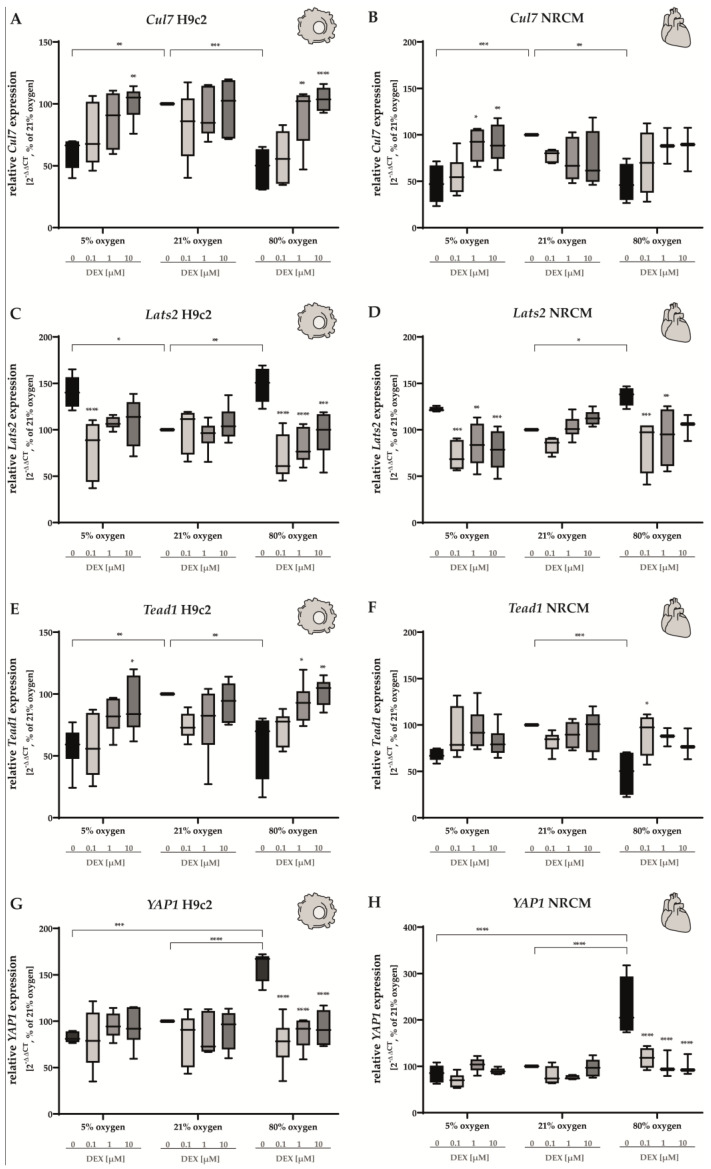
Expression of (**A**,**B**) *Cul7*, (**C**,**D**) *Lats2*, (**E**,**F**) *Tead1*, and (**G**,**H** *YAP1* of H9c2 cells (left column) and of NRCMs (right column) was determined for normoxia (21% O_2_), hypoxia (5% O_2_) or hyperoxia (80% O_2_). Quantitative analysis was conducted for the DEX-treated cells with the concentrations 0.1 µM (light gray bars), 1 µM (gray bars) and 10 µM (dark gray bars) in comparison to the untreated control group (black bars). Data are normalized to the level of cells exposed to normoxia (21% O_2_) at each time point (control 100%, black bars). n = 6 individual experiments/group. * *p* < 0.05, ** *p* < 0.01, *** *p* < 0.001, **** *p* < 0.0001 (ANOVA, Bonferroni’s post hoc test).

**Table 1 antioxidants-12-01206-t001:** Sequences of oligonucleotides.

	Oligonucleotide Sequence 5′−3′	Accession No.
	AIF	
forward	CACAAAGACACTGCAGTTCAGACA	NM_031356.1
reverse	AGGTCCTGAGCAGAGACATAGAAAG	
probe	6-FAM-AGAAGCATCTATTTCCAGCC-TAMRA	
	Atg5	
forward	ACATCAGCATTGTGCCCCA	NM_001014250.1
reverse	TGTCATGCTTCGGTGTCCTG	
probe	6-FAM-CAGACTGAAGGCCGTGTCCTGCTCA-TAMRA	
	Atg12	
forward	TCTGCCTAGCCTGGAACTCAG	NM_001038495.1
reverse	TAGCCCTGTGTGCTCTGCTTT	
probe	6-FAM-CCTGTCCGTGAAGCTCACCCAGC-TAMRA	
	Casp3	
forward	ACAGTGGAACTGACGATGATATGG	NM_012922.2
reverse	AATAGTAACCGGGTGCGGTAGA	
probe	6-FAM-ATGCCAGAAGATACCAGTGG-TAMRA	
	Casp8	
forward	GGACTATCCTGGCAGAAAAC	NM_022277.1
reverse	TCACCTCATCCAAAACAGAAAC	
probe	6-FAM-AGGATCGACGATTACGAACGATCAAGCACA-TAMRA	
	Cul7	
forward	GATCCTTCTGTCACTGAGCC	XM_032900620.1
reverse	TCCCAGCATTCAACTCCTCC	
probe	6-FAM-CCGCTGCGCCCTGCTTGCACT-TAMRA	
	CycD2	
forward	CGTACATGCGCAGGATGGT	NM_199501.1
reverse	AATTCATGGCCAGAGGAAAGAC	
probe	6-FAM-TGGATGCTAGAGGTCTGTGA-TAMRA	
	GCLC	
forward	GGAGGACAACATGAGGAAACG	NM_012815.2
reverse	GCTCTGGCAGTGTGAATCCA	
probe	6-FAM-TCAGGCTCTTTGCACGATAA-TAMRA	
	Hif1α	
forward	GCGCCTCTTCGACAAGCTT	NM_024359.2
reverse	CTGCCGAAGTCCAGTGATATGA	
probe	6-FAM-AGAGCCCGATGCCCTGACTCTGCT-TAMRA	
	Lats2	
forward	ACTACCAGAAAGGGAACCAC	NM_001107267.1
reverse	CAAAAAGAATCACACCGACAC	
probe	6-FAM-CTGGTGACCTCTGGGACGACGTTTCCAA-TAMRA	
	Nrf2	
forward	ACTCCCAGGTTGCCCACAT	NM_031789.2
reverse	GCGACTCATGGTCATCTACAAATG	
probe	6-FAM-CTTTGAAGACTGTATGCAGC-TAMRA	
	TATAbp	
forward	CACCGTGAATCTTGGCTGTAAAC	NM_001004198
reverse	CGCAGTTGTTCGTGGCTCTC	
probe	6-FAM-TCGTGCCAGAAATGCTGAATATAATCCCAA-TAMRA	
	Tead1	
forward	AAACTGAGGACGGGAAAGAC	NM_001198589.2
reverse	AGACGATCTGGGCTGATGAC	
probe	6-FAM-ACAAGCATGGATCAGACTGCCAAGGACAA-TAMRA	
	Timp1	
forward	CGGACCTGGTTATAAGGGCTAA	NM_053819.1
reverse	CGTCGAATCCTTTGAGCATCT	
probe	6-FAM-AGAAATCATCGAGACCACCT-TAMRA	
	Timp2	
forward	GGCAACCCCATCAAGAGGAT	NM_021989.2
reverse	GGGCCGTGTAGATAAATTCGAT	
probe	6-FAM-AGATGTTCAAAGGACCTGAC-TAMRA	
	Tnnt2	
forward	GCGAAGAAGAGGAAGACGAG	NM_012676.1
reverse	CACCAAGTTGGGCATGAAG	
probe	6-FAM-CAGTAGAGGACTCCAAACCCAAGCCCAGCA-TAMRA	
	YAP1	
forward	TGCTGCTCAACATCTCAGAC	NM_001394328.1
reverse	TGCTCCCATCCATCAGGAAG	
probe	6-FAM-CCCGGAAGGCCATGCTCTCCCAACTGAA-TAMRA	

Abbreviations: apoptosis-inducing factor (*AIF*); autophagy-related 5/12 (*Atg5/12*); caspase−3/8 (*Casp3/8*); cullin 7 (*Cul7*); cyclin D2 (*CycD2*); glutamate-cysteine ligase catalytic subunit (*GCLC*); hypoxia-inducible factor 1α (*Hif1α*); large tumor suppressor kinase 2 (*Lats2*); nuclear factor-erythroid 2-related factor 2 (*Nrf2*); TATA-binding protein (*TATAbp*); TEA domain transcription factor 1 (*Tead1*); tissue inhibitor of metalloproteinases 1/2 (*Timp1/2*); troponin T2, cardiac type (*Tnnt2*); Yes-associated protein 1 (*YAP1*); *6*-*carboxyfluorescein* (6-FAM); and tetramethylrhodamine (TAMRA).

## Data Availability

The data used to support the findings of this study are available from the corresponding author upon request. The analyzed data used to create the graphs and statistical evaluation are attached in the supplemented material of this work.

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
