# Peer review of "Cardioprotective Effects of Dexmedetomidine in an Oxidative-Stress In Vitro Model of Neonatal Rat Cardiomyocytes"

_antioxidants, 2023, doi:10.3390/antiox12061206_

Round 1

Reviewer 1 Report

What do authors mean by “oxidative stress” here? Both hypoxia and hyperoxia are considered oxidative stress? If “oxidative stress” means “ROS stress”, please provide evidence of the role of ROS in hypoxia and hyperoxia in their models.

In our body, cells do not normally see 21% O2. In this regard, perhaps moderate O2 increased cell proliferation and high O2 decreased proliferation? Regardless, DEX did not really do anything to moderate or high O2-mediated changes in cell proliferative states.

There are too many references.

Reviewer 2 Report

This manuscript describes the use of cultured cardiomyocytes to look at the effects of hypoxia and hyperoxia on cardiomyocyte development, and it also examines the use of the drug Dexmedetomidine (an inhibitor of reactive oxygen species) on cell survival under different oxygen levels.  This study shows that inhibiting the formation of reactive oxygen species with DEX improved cell survival and helped maintain expression of genes involved in cell growth. 

This is a well written paper and nicely designed study.  Can the authors site papers or report what levels of oxygen are seen by the heart during development (high vs low) and how this relates to the levels of oxygen used in the study?

Also, could the authors please comment on some of the other factors that may contribute to increases in oxidative stress in the prenatal heart?

Reviewer 3 Report

The Abstract is poorly written. Understanding Dexmedetomidine's (DEX) cardioprotective effects under varied oxidative situations is exceedingly challenging. According to the authors' Abstract, hypoxia and hyperoxia considerably affected cardiomyocyte proliferating and degenerating processes. However, it is not specified how Dexmedetomidine impacts all these processes or which biomarkers DEX alters. Therefore, using the findings, the Abstract should be revised.

The cardioprotective properties of DEX in oxidative conditions were assessed using hypoxia and hyperoxia conditions. Most of the time, authors exclusively discuss DEX's cardioprotective effects in hypoxia and hyperoxia. But the results and discussion sections need an interpretation, a scientific justification, and the correct citation of the earlier research.

Only the authors presented the gene expression of several biomarkers associated with apoptosis, autophagy, and the Hippo YAP pathway. Numerous scientific explanations claim that gene modification is unrelated to the disease's onset. Respective protein expression is crucial to describing the disease condition, and it's critical to demonstrate how medications, phytochemicals, and synthetic compounds modify protein expression in diverse disease conditions to explain how well they work to treat the specific disease.

Last but not least, it is not entirely evident throughout the text how DEX demonstrated cardioprotective benefits on newborn rat cardiomyocytes under oxidative conditions.  

Minor modification is needed

Round 2

Reviewer 1 Report

.

Reviewer 3 Report

The authors provide a satisfactory answer to most queries, and the revised version of this manuscript is well-improved.